

# Cancer prevention recommendations: awareness in a Mexican public hospital

Alejandro Trujillo Rivera[1], Clara Luz Sampieri[2], Eduardo Antonio Trujillo Rivera[3] and José Roberto Gómez Cruz[4]

[1] Instituto Nacional de Cancerología, Ciudad de México, Mexico
[2] Instituto de Salud Pública, Universidad Veracruzana, Xalapa, Veracruz, Mexico
[3] Department of Pediatrics, Division of Critical Care Medicine, Children's National Health System, Washington, District of Columbia, United States
[4] Centro de Alta Especialidad "Dr. Rafael Lucio", Xalapa, Veracruz, Mexico

Corresponding author
Clara Luz Sampieri,
csampieri@uv.mx

## ABSTRACT

**Background:** The recommendations of both the World Cancer Research Fund (WCRF) and the American Institute for Cancer Research (AICR) for the prevention of cancer are important public health tools. These recommendations for the prevention of specific cancers are related to body weight maintenance; physical activity; foods and drinks that promote weight gain; plant foods; foods of animal origin; alcoholic beverages; preservation, processing, and preparation of food; food supplements; and breastfeeding.

**Methods:** This study was a descriptive, cross-sectional, retrospective study. All patients provided written informed consent prior to enrollment in the study. Stratified random sampling was carried out with a convenience sample size of ≥280 participants. The characteristics of the participants were investigated using validated questions. Knowledge about the WCRF/AICR recommendations for the primary prevention of cancer was determined using 14 multiple choice questions validated in this study. Group A included participants who answered that cancer can be prevented and that lifestyle is the main factor related to the onset of cancer; the remaining participants were assigned to Group B. The χ2 test and Mann–Whitney U test were used to determine differences in the groups. A $p$-value of ≤0.05 was considered statistically significant. A multiple linear regression analysis with gamma response was performed to assess the knowledge score.

**Results:** A total of 289 participants were included; 168 (58%) participants were in group A, and 121 (42%) participants were in group B. Using a 0 to 14 scale, the median (P25, P75) number of correct answers was 11 (10, 12). There was no evidence of a difference between groups in sex, relationship status, literacy skills, years of education, occupation, monthly income per person, and BMI. Most of the participants reported that they did not consume tobacco ($n = 259/289$) or alcohol ($n = 261/289$) in the week prior to completing the survey.

**Discussion:** Most of the participants (58%) considered cancer preventable and agreed that lifestyle factors were the most important factors in cancer development. The results also showed a high level of public awareness of some evidence-based associations between cancer and lifestyle factors, such as tobacco use. Some confusion among the public on other risk factors was also identified: waist circumference, a variable related to excess weight, was not identified as a risk factor for cancer by most of the participants, but the consumption of foods and beverages

high in calories was identified as a risk factor by the majority of participants. Awareness of cancer protective and risk factors may lead to positive behavioral changes and eventually reduce the burden of cancer.

# INTRODUCTION

Both the World Cancer Research Fund (WCRF) and the American Institute for Cancer Research (AICR) have argued that tobacco, diet, body weight, and physical activity are important factors related to the development of cancer. The WCRF/AICR have released reports on cancer prevention that emphasize the importance of research on this topic for public health. These reports cover cancers of the mouth, pharynx, larynx, nasopharynx, esophagus, lung, stomach, pancreas, gallbladder, liver, colon, rectum, breast, ovary, endometrium, prostate, kidney, skin, bladder, and cervix (*CUP, 2018*). According to the WCRF, high percentages of body fat in adults can cause cancer, and diets containing greater amounts of processed foods high in fat, starches, or sugars are causes of weight gain, overweight, and obesity. WCRF states that "consumption of either red or processed meat are both causes of colorectal cancer"; physical activity protects against colon, breast, and endometrial cancers; and lactation protects the mother against breast cancer. So, the WCRF/AICR recommendations for the prevention of specific cancers are related to body weight maintenance; physical activity; foods and drinks that promote weight gain; plant foods; foods of animal origin; alcoholic beverages; preservation, processing, and preparation of food; food supplements; and breastfeeding (*CUP, 2018*).

In 2021, cancer was the third leading cause of death in Mexico, accounting for 90,123 registered deaths and 8% of the total number of deaths (*INEGI, 2023a, 2023b*). The states with the highest cancer death rates were Mexico City, Colima, and Veracruz, with 9.3, 8.5, and 8.1 cancer deaths per 10,000 habitants, respectively (*INEGI, 2023a*). However, according to Brau-Figueroa, in Mexico "the real incidence and prevalence" of cancer is unknown (*Brau-Figueroa, Palafox-Parrilla & Mohar-Betancourt, 2020*).

In Mexico, an estimated 191,000 new cases of cancer are diagnosed annually (*AMLCC, 2023*). A concordance analysis of selected WCRF/AICR recommendations for cancer prevention, which included 790 Mexicans aged 18–70 years, found that no participants adhered to the recommendations (*Vossenaar et al., 2010a*). In the same Mexican population, it was reported that no participants met the WCRF/AICR recommendations of body mass index (BMI) or daily intake of refined sugar (*Vossenaar et al., 2010b*). A subsequent analysis revealed that 77% of the participants complied with at least half of the eight dietary and eight lifestyle WCRF/AICR recommendations for cancer prevention (*Vossenaar et al., 2011*). However, no studies on the Mexican population have examined knowledge of the WCRF/AICR recommendations for cancer prevention and related behaviors, such as tobacco use and alcoholic beverages. Knowledge is not enough to change behavior, but it is a key component (*Yu & Baade, 2020*), as increased knowledge

has the potential to motivate behavioral change (*Arlinghaus & Johnston, 2018*). Thus, the aim of this study was to evaluate the level of knowledge about WCRF/AICR cancer prevention recommendations and identify participants' tobacco and alcohol use in the week prior to completing the survey.

## MATERIALS AND METHODS

This study was approved by the Ethics Committee of the public hospital Centro de Alta Especialidad "Dr. Rafael Lucio," Institutional Review Board (IRB approval number: 01/18). All patients provided written informed consent prior to enrollment in the study. A descriptive, cross-sectional, retrospective study with minimal risk was conducted. Through face-to-face interviews, data were collected using validated questions (see below). Trained personnel measured each participant's body weight and height. Participants were recruited from patients who had outpatient consultations from the health services, psychology, nutrition, rehabilitation, or dentistry departments of Centro de Alta Especialidad "Dr. Rafael Lucio" located in Xalapa, Veracruz, Mexico. Patients had to be 18 years of age or older and agree to participate in the study by signing an informed consent form to be included. The single exclusion criterion was a diagnosis of cancer. The sampling method consisted of randomly selecting five clinics and subsequently choosing five or 10 patients, at random, from the morning shift of appointments at the selected clinics. A convenience sample size of ≥280 participants was established.

For body weight determination, the participants wore light clothing with empty pockets and no shoes and were measured to the closest 100 g on calibrated scales. The Body Mass Index (BMI) criteria used to classify participants were based on those outlined in the WCRF/AICR report (*CUP, 2018*). Interviews and measurements were carried out in February 2018. The demographic characteristics of the participants (sex, relationship status, literacy, years of education, occupation, and monthly income per person) were assessed using questions validated in a previous study (*Trujillo Rivera et al., 2018*).

Knowledge about the WCRF/AICR recommendations for the primary prevention of cancer (*CUP, 2018*) was determined using 14 multiple choice questions with the following three response options: "Yes," "No," or "I don't know." Tobacco use and alcoholic beverage consumption in the week prior to completing the survey was determined using multiple choice questions with the following response options: "Always," "Almost always," "Sometimes," "Almost never," or "Never." The questions designed for this study about WCRF/AICR recommendations underwent content validation by three experts, and a pilot test ($n = 20$) was carried out to detect errors or omissions (File SA).

The independent variables were sex, relationship status, literacy skills, years of education, occupation, monthly income per person, BMI, tobacco use in the week prior to the survey, and alcoholic beverage consumption in the week prior to completing the survey. The dependent variable—knowledge about the primary prevention of cancer consistent with the WCRF/AICR recommendations—was calculated using the answers to the 14 questions, with one point assigned to correct answers and zero points assigned to incorrect answers or responses of "I don't know" (hereafter, this point value is referred to as the knowledge score). Sex was classified as either female or male. Relationship status was

classified as having a partner (married or unmarried) or without a partner (single, widowed, or divorced). Years of education was divided into ≤11 years or ≥12 years, with ≥12 years of education considered equivalent to the participant achieving at least secondary education. Monthly income per person was classified as low income (<MXN 2400.00) or high income (≥MXN 2400.00; CONEVAL, 2019). In 2018, the average exchange rate was USD 1 = MXN 19.23. BMI was classified into two groups with the first including underweight (<18.5 kg/m$^2$) and normal weight (18.5–24.9 kg/m$^2$) and the second including overweight (25.0–29.9 kg/m$^2$) and obesity (30.0 to ≥40 kg/m$^2$). Responses to questions about tobacco use and alcoholic beverage consumption in the week prior to completing the survey were dichotomized as "Yes" ("Always," "Almost always," "Sometimes," and "Almost never") or "No" ("Never").

Groups were formed according to the answers to questions 17 and 18 (File SA). Group A included the participants who answered that cancer can be prevented and that lifestyle is the main factor related to the onset of cancer. The remaining participants were assigned to Group B. The distributions of variables were determined, and the totals, percentages, medians, and percentiles (25th (P25) and 75th percentiles (P75)) were calculated. The χ2 test was used to determine if there were group differences in the percentages, and the Mann–Whitney U test (MW-U test) was used to determine if there were group differences in the distributions of the continuous variables. A $p$-value of ≤ 0.05 was considered statistically significant.

A multiple linear regression analysis with gamma response was performed to assess the knowledge score (range: 0–14). The model included dichotomous covariates (sex, occupation, group, monthly income per person, and years of education) and a continuous covariate (age). For the multiple linear regression analysis, generalized linear models with a gamma distribution were used. All linear models with interactions were considered. The first model used to describe the data was chosen using the Akaike Information Criterion (AIC). The best-fitting model was selected according to AIC value and by consecutively eliminating the variables that were not statistically significant in a likelihood ratio test. The chi-square test of deviance of the model resulted in a $p$ value > 0.1, and residual plots did not show concerning patterns. Ratios of the mean scores between the groups were calculated using the contrasts of the coefficients of the best-fitting model. StatCal version 7 (Epi Info$^{TM}$) and R (version 4.3.0 (R Core Team, 2023) "Already Tomorrow") were used to perform the analyses.

## RESULTS

A total of 289 people participated in this study, the majority of whom were women ($n$ = 217, 75%). The response rate was 100%. The median (P25, P75) age of the participants was 41 years (29, 55), and the median monthly income per person was MXN 1,000 (600; 1,762). Most of the women reported being homemakers ($n$ = 126/217, 58%), and only two men out of 72 (3%) indicated that they were homemakers.

Group A included 168 people and Group B included 121. There was no evidence of a difference between the groups in median (P25, P75) age (Group A: 40 (26, 51); Group B: 43, (31, 60); $p$ = 0.062, MW-U test) or monthly income per person (Group A: MXN 1,000

**Table 1 General characteristics of the study group.**

| Characteristic | Group A (n = 168) | | Group B (n = 121) | | p-value |
|---|---|---|---|---|---|
| **Sex** | n | % | n | % | |
| Woman | 125 | 74 | 92 | 76 | 0.859 |
| Man | 43 | 26 | 29 | 24 | |
| **Having a partner** | | | | | |
| No | 61 | 36 | 49 | 40 | 0.548 |
| Yes | 107 | 64 | 72 | 60 | |
| **Literacy skills** | | | | | |
| No | 14 | 8 | 6 | 5 | 0.379 |
| Yes | 154 | 92 | 115 | 95 | |
| **Years of education** | | | | | |
| ≤11 | 127 | 76 | 84 | 69 | 0.302 |
| ≥12 | 41 | 24 | 37 | 31 | |
| **Occupation** | | | | | |
| Housework | 69 | 41 | 59 | 49 | 0.239 |
| Others | 99 | 59 | 62 | 51 | |
| **Monthly income per person** | | | | | |
| Low income | 143 | 85 | 105 | 87 | 0.820 |
| High income | 25 | 15 | 16 | 13 | |
| **BMI** | | | | | |
| Under weight and normal | 67 | 40 | 49 | 40 | 0.999 |
| Overweight and obesity | 101 | 60 | 72 | 60 | |

**Notes:**
Group A: participants who answered that cancer is preventable through lifestyle.
Group B: participants who chose other options.
BMI: Body mass index.

(612, 1,683); Group B: MXN 1,000 (500, 1,875); $p = 0.457$, MW-U test). There was no evidence of a difference between the groups in sex, relationship status, literacy skills, years of education, occupation, or BMI (Table 1).

Most of the participants reported that they did not smoke tobacco in the week prior to completing the survey ($n = 259$), with no evidence of a difference between the groups (Group A: $n = 149$, (87%); Group B: $n = 110$ (90%); $p = 0.323$). Similarly, most of the sample reported that they did not consume alcoholic beverages in the week prior to completing the survey ($n = 261$), with no evidence of a difference between the groups (Group A: $n = 154$ (92%); Group B: $n = 107$ (88%); $p = 0.607$). A total of 90 participants (31%) were not overweight, did not smoke the week prior to completing the survey, and did not drink alcoholic beverages in the week prior to completing the survey.

The marginal median (P25, P75) knowledge score was 11 (10, 12), with men scoring higher (12 (10, 13)) than women (11 (9, 12); $p = 0.004$, MW-U test). Compared with other occupations (12 (10, 13)), being a homemaker was associated with a lower knowledge score (11 (9, 12); $p = 0.019$, MW-U test). People aged ≤ 40 years had higher knowledge scores (12 (10, 13)) than those aged < 40 years (11 (9, 12); $p = 0.008$, MW-U test). There were no

**Table 2 Knowledge of cancer prevention, as measured by number of correct answers.**

| Topics of the questions related to the prevention of cancer | Group A (n = 168) | | Group B (n = 121) | | Percentages comparisons[2] | |
|---|---|---|---|---|---|---|
| | Correct answers | | | | | |
| | n | % (95% CI)[1] | n | % (95% CI)[1] | Differences (A–B) (95% CI) | p-value |
| Waist circumference | 80 | 47.6 [39.9–55.5] | 35 | 28.9 [21–37.9] | 18.7 [7.1–29.4] | 0.002 |
| Excess weight | 106 | 63.1 [55.3–70.4] | 56 | 46.3 [37.2–55.6] | 16.8 [4.8–28.2] | 0.006 |
| Smoking tobacco | 159 | 94.6 [90.1–97.5] | 118 | 97.5 [92.9–99.5] | −2.9 [−7.9 to 2.3] | 0.362 |
| Sedentary lifestyle | 157 | 93.5 [88.6–96.7] | 116 | 95.9 [90.6–98.6] | −2.4 [−8.2 to 3.5] | 0.532 |
| Physical activity | 133 | 79.2 [72.2–85] | 89 | 73.6 [64.8–81.2] | 5.6 [−4.5 to 15.9] | 0.320 |
| Consumption of foods with high caloric densities | 159 | 94.6 [90.1–97.5] | 109 | 90.1 [83.3–94.8] | 4.6 [−1.7 to 11.9] | 0.214 |
| Consumption of high-calorie drinks | 163 | 97 [93.2–99] | 117 | 96.7 [91.8–99.1] | 0.3 [−4.3 to 5.8] | 0.875 |
| Consumption of vegetables and fruits | 149 | 88.7 [82.9–93.1] | 108 | 89.3 [82.3–94.2] | −0.6 [−8 to 7.6] | 0.880 |
| Consumption of red meat | 121 | 72 [64.6–78.7] | 77 | 63.6 [54.4–72.2] | 8.4 [−2.8 to 19.5] | 0.166 |
| Consumption of processed meat | 142 | 84.5 [78.2–89.6] | 92 | 76 [67.4–83.3] | 8.5 [−0.8 to 18.2] | 0.096 |
| Consumption of alcoholic beverages | 132 | 78.6 [71.6–84.5] | 80 | 66.1 [57–74.5] | 12.5 [2–23.2] | 0.026 |
| Consumption of foods high in salt | 151 | 89.9 [84.3–94] | 112 | 92.6 [86.3–96.5] | −2.7 [−9.6 to 4.6] | 0.564 |
| Consumption of food supplements | 119 | 70.8 [63.3–77.6] | 90 | 74.4 [65.6–81.9] | −3.5 [−13.8 to 7.5] | 0.595 |
| Breastfeeding | 116 | 69.0 [61.5–75.9] | 82 | 67.8 [58.7–76] | 1.3 [−9.7 to 12.4] | 0.918 |

Notes:
Group A: participants who answered that cancer is preventable through lifestyle.
Group B: participants who chose other options.
[1] Confidence intervals computed using exact methods for binomial distributions.
[2] Unconditional exact test and confidence intervals for difference of proportions between group A and group B.

evidence of a difference in the knowledge scores by relationship status (11 (10, 12) for those reporting "having a partner" and those "without a partner"; $p = 0.873$, MW-U test), years of education (11 (9, 12) *vs.* 12 (10, 13); $p = 0.133$, MW-U test), monthly income per person (11 (10, 12) *vs.* 11 (10, 13); $p = 0.906$, MW-U test), and BMI classification (overweight/obesity: 11 (10, 13) *vs.* underweight/normal weight: 11 (10, 12); $p = 0.234$, MW-U test).

There was evidence for difference in knowledge scores between the groups (Group A: 12 (10, 13); Group B: 11 (9, 12); $p = 0.004$). In both groups, more than 90% of the participants reported that the consumption of foods and beverages with high caloric densities was related to the development of cancer, but fewer participants were aware of the recommendations regarding waist circumference and weight (Table 2). Group A had higher knowledge scores on the questions about cancer risks related to high waist circumference, excess weight, and the consumption of alcoholic beverages ($p < 0.05$), but there was no evidence of a difference between the groups on all other questions (Table 2).

A comparison of the knowledge scores revealed no significant sex differences after adjusting for group, monthly income per person, and years of education (File SB). The results also indicated that among those who were not homemakers, older age was associated with higher knowledge scores regarding the WCRF/AICR recommendations. Specifically, a 10-year increase in age was associated with a 3.9% (95% CI [1.8–6.1%]) increase in the mean knowledge score, a 20-year increase in age was associated with an

8.1% (95% CI [3.7–12.8%]) increase in the average knowledge score, and a 40-year increase in age was associated with an increase of 17.0% (95% CI [7.5–27.1%]). Using the appropriate contrasts, a relationship between age and knowledge scores was not found among those not involved as homemakers (0.2%; 95% CI [0.0–0.3%]; $p$ = 0.06).

## DISCUSSION

In this survey, most of the participants were aware of the majority of WCRF/AICR recommendations for the primary prevention of cancer, except for those pertaining to waist circumference and excess weight. The results of this study are difficult to compare with those of other studies because other studies have included younger samples. For example, among undergraduate students in Nigeria, the recommendation for cancer prevention regarding waist circumference was also one of the least known guidelines (58.2%), however, maintaining a healthy weight was identified by most participants (80.2%; *Folasire, Folasire & Chikezie, 2016*). A study of adolescents in the United Kingdom reported low knowledge about cancer (*Kyle, Forbat & Hubbard, 2012*).

The results of this study indicated that most of the participants considered cancer preventable with lifestyle factors being the most crucial factors for cancer development. These findings are consistent with those reported in Omani adolescents (*Al-Azri et al., 2019*).

In Denmark, a strong socioeconomic gradient in cancer awareness has been reported (*Hvidberg et al., 2014*), but the current study did not observe associations between knowledge scores and years of education or personal monthly income, likely because the included population was highly vulnerable: they had low education levels and low incomes. In Mexico, in 2021, 26.5% of the population lacked affiliation with health services, and in 2022, 38.3% had incomes lower than the monetary value of a food basket (*CONEVAL, 2022*). This is important context because "persistent poverty" has been associated with cancer mortality through factors such as patterns of risk behaviors, health care access, infrastructure, and social determinants of health (*Moss et al., 2020*). Consistent with the findings of other studies, this study found a high level of public awareness of some evidence-based associations between cancer and lifestyle factors, such as tobacco use (*Kabalan et al., 2021*). There seemed to be some confusion among participants, however, because the consumption of foods and beverages high in calories was identified by the majority of participants as a risk factor for cancer, but waist circumference, a variable related to excess weight, was not identified as a risk factor for cancer by most of the participants. It is necessary to raise awareness among the population that waist circumference is an indicator associated with diet and physical activity.

The results of this study also indicated that people who were homemakers, and those aged 18 to 26, had lower knowledge scores than those who were not homemakers. The knowledge scores of those who were homemakers did not tend to increase with age. There are no previous studies on the knowledge of cancer prevention in people who are homemakers. In Mexico, in 2021, it was estimated that 84 million people performed unpaid homemaking activities and that 2.2 million people engaged in paid housework activities (*CONEVAL, 2022*).

There are limitations of this study. All conclusions in this study were based on interpretations of statistical tests based on hypotheses and on the developed statistical model. The size of the sample may also have been a limitation in detecting important, but slight differences in the knowledge scores of subgroups. Although the hospital population included in this analysis was intended to be representative of an entire region, the data were applicable only to the study population and are not representative. The present survey is one of the few studies that has aimed to evaluate cancer prevention knowledge consistent with the WCRF/AICR and cancer-related lifestyle behavior recommendations among the adult Mexican population. Social inequalities in Mexico mainly affect the most vulnerable populations, such as women. According to Flamand and collaborators, "although it is usually thought that risk factors for cancer are habits freely chosen, most are related to social and economic conditions that are not chosen" (*Flamand, Moreno-Jaimes & Arriaga-Carrasco, 2020*). Thus, the probability of developing cancer depends on social variables such as ethnicity, gender, and occupation. Moreover, diagnosis and treatment depend on educational and economic factors and access to health services (*Flamand, Moreno-Jaimes & Arriaga-Carrasco, 2020*).

## CONCLUSIONS

This study was carried out in a vulnerable population, as most of the respondents were women with low incomes and low educational levels. The results showed that homemakers do not increase their knowledge about cancer prevention with age. In Mexico, it has been documented that women who are homemakers perceive themselves as socially isolated and are the population most impacted by certain communicable diseases. Implementing public and political interventions for cancer control and prevention, while considering the sociodemographic characteristics of the population, is important as awareness of cancer protective and risk factors may lead to positive behavioral changes and eventually reduce the burden of cancer in Mexico. To reduce the burden of cancer in the population, preventive measures should emphasize the potential of engaging in low-risk healthy behavior patterns rather than specific risk factors. The WCRF/AICR recommendations are important tools in this effort as they include simple, clear messages aimed at the public.

## ACKNOWLEDGEMENTS

We would like to thank all participants of this study.

### Funding

The authors received no funding for this work.

### Competing Interests

The authors declare that they have no competing interests.
## Author Contributions

- Alejandro Trujillo Rivera conceived and designed the experiments, performed the experiments, analyzed the data, authored or reviewed drafts of the article, and approved the final draft.
- Clara Luz Sampieri conceived and designed the experiments, performed the experiments, analyzed the data, prepared figures and/or tables, authored or reviewed drafts of the article, and approved the final draft.
- Eduardo Antonio Trujillo Rivera analyzed the data, authored or reviewed drafts of the article, and approved the final draft.
- José Roberto Gómez Cruz conceived and designed the experiments, analyzed the data, authored or reviewed drafts of the article, and approved the final draft.

## Human Ethics

The following information was supplied relating to ethical approvals (*i.e.*, approving body and any reference numbers):

The study was approved by the Ethics Committee of the public hospital Centro de Alta Especialidad "Dr. Rafael Lucio", Institutional Review Board (01/18).

## Data Availability

The participants' scores are available in the Supplemental Files.

## Supplemental Information

Supplemental information for this article can be found online at http://dx.doi.org/10.7717/peerj.17593#supplemental-information.

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
