# Peer review of "Cancer prevention recommendations: awareness in a Mexican public hospital"

_PeerJ, doi:10.7717/peerj.17593_

## Round 0.1 · original submission · Major Revisions

Thank you for your patience. I am not your original editor but have been invited to take over your manuscript, which looks at an interesting question. We have recommendations from two reviewers, who you will note have quite different views of your work. I’ve made my own review of your manuscript (with my comments below) and will invite you to respond to the suggestions from our reviewers and myself. I feel that substantial work will likely be needed to address these. If you are willing to address these remarks, then for each point, please provide a comment indicating either how you have changed your manuscript in response (with these changes tracked) or why you do not believe changes should be made. While the reviewers have not raised a large number of points collectively, I feel that each of their point raised warrants consideration and response. I look forward to seeing your revised manuscript in due course.

Abstract: I agree with Reviewer 1 that the abstract needs attention to identify what information is appropriate to include here (e.g., ethics approval is important to document in the manuscript but doesn’t usually warrant mention in the abstract), ensure clarity (e.g., the sample size, whether this is a target or the number achieved, cannot be an inequality, the two sample sizes cannot both be 168 for a total of 289 on Lines 38–39), and make its structure appropriate (e.g., some of the background section seems to be methods, see also the Reviewer’s comments).

Lines 29–30: While I appreciate that the manuscript was edited for language by another publisher, this sentence is not grammatically correct: “The characteristics of the participants were assessed using questions validated.” There are other instances where the writing needs attention. The meaning is clear so you can leave this to address after another round of reviews if you prefer.

Line 109: The stratifying variable(s) need to be identified here. I’m not sure what “considering the number of participating clinics” is intended to communicate here. If the strata are clinics, this would seem clustered but perhaps proportional to size (measured in patients)? In any case, the details need to be made clear as this has implications for analyses. If the data is clustered by clinics, then this adds a random effect or similar to the analyses.

Line 110: This reads as if 280 was determined as an appropriate number of patients to recruit. How was this number arrived at? If this is based on some n per variable, pragmatic, or other criterion, please describe this here.

Lines 148: I don’t see how readers would appreciate what these questions are at this point (particularly since these appear to be Q17 and Q18 in the supplement with the questions). Based on my reading here, two ‘correct responses would go in Group A. Whereas one ‘correct and one ‘incorrect, or two ‘incorrect responses would go in Group B. This seems to make Group B more heterogeneous than A. Why not an intermediate group for one correct and one incorrect, for example?

Line 156: Multivariate would mean multiple measurements on the unit of analysis (here, people, although if there is clustering by clinics, this label would be appropriate). I think you mean multivariate given the description of the analyses at this stage. However, this invites the question of why non-parametric methods were used for univariable comparisons of the two groups (Line 153) but a parametric model was considered appropriate for the multivariable model of score. What aspects of the variables and/or model diagnostics led to these decisions? How do you know that a gamma distribution was appropriate? Also using regression approaches in the univariable cases would add value through allowing a difference and 95% CI rather than just a p-value. Table 2 would be much more informative with this information.

Lines 157–158: Some of these variables seem plausibly on the causal pathway (i.e., could mediate the association of interest), e.g. sex -> score could include occupation, income, and education on the causal pathway/as mediators, could it not? Similarly, age -> score seems likely to include at least income on the causal pathway. Are you interested in total or direct effects here? A DAG might be useful to the reader to understand your thinking about the underlying causal model here.

Lines 160–163: As Frank Harrell has explained in his excellent Regression Modeling Strategies, using AIC is equivalent to backwards elimination and causes the same problems as any other data-driven variable selection strategy (including LR tests). I recommend that you reconsider this approach but I’m happy to consider your justification for it.

Line 170: This seems an extraordinary response rate for people to provide informed consent to participate. Are you sure that no one was invited and declined to take part?

Lines 174–175: “no difference” is not the same as there being “insufficient/no evidence of a difference”.

Lines 169–170: With 217/289 being women, each person contributes 0.35 of a percent, making the second decimal place uninformative and the first one not entirely meaningful. Personally, I’d use integer percentages here, or at least no more than one decimal place.

Line 182: Be careful with the number of decimal places for p-values.

Line 189: Is this p-value from gamma regression or a MW-U test? Similar for the following. The statistical methods refer to MW-U for comparing groups, which here could be interpreted as Group A and Group B only.

Table 1: It’s unclear why you’ve broken down obesity categories into 3 given the small numbers in the higher categories.

Supplementary table: It is unclear why there are coefficients for both being male and for being female.

Reviewer 1 ·

Basic reporting

1. Line 39: gourp B (168, 58.13%) may be incorrect. (121, 41.87%)?
2. In abstract, there may be concise conclusions. This paper includes a section of discussion in the abstract, which seems should be in the main text. Authors are recommended to read the instructions for authors at the journal website to make sure what should be included in the abstract.

Experimental design

This study might be somewhat interesting, but its significance is limited. While it investigated some issues related to awareness of cancer prevention, there isn't enough data to indicate whether this awareness, or lack thereof, translates into certain impacts on individuals, such as dietary choices, nutrient intake, physical activity, and so on. There were notable differences in the accuracy of answers between Groups A and B in a few questions, but from the data, we cannot further understand whether these cognitive differences might result in behavioral or physiological changes (e.g., waist circumference, BMI, alcohol consumption).
Moreover, for this type of cross-sectional study, the sample size of 280 people raises doubts about its representativeness. The authors label this study as prospective, which confuses me because I didn't see any follow-up or data demonstrating causality. Unless the authors transform this study into a long-term observational cohort of these individuals, with cancer awareness as exposure and the development of cancer as an outcome, conducting years-long observations to establish causal links, then it might qualify as a prospective study. However, this would likely be challenging to execute, and with such a small sample size, it would be difficult to observe a sufficient number of outcomes.

Validity of the findings

no comment

·

Basic reporting

The article is well written, with adequate use of the English language and with a satisfactory theoretical foundation.
Two points must be considered by the authors:
Lines 59 – 63:
It is not clear what the recommendations are for preventing cancer. For example, does weight gain prevent cancer? Consumption of alcoholic beverages prevents cancer... I suggest the authors rewrite the paragraph more clearly.
Lines 66 - 68; 70 -73; 80 - 82...
Authors must check the citation format. Is the appropriate reference like the ones highlighted in these lines or the AUTHOR/DATE format?

Experimental design

The study meets methodological recommendations. Authors should only clarify the following point:
Lines 148 - 150
Checking the questionnaire available in supplementary material A, questions 15 and 16 do not match the highlighted questions. Please check.

Validity of the findings

Adequate

Additional comments

I am grateful for the opportunity to review the article "Cancer prevention recommendations: awareness in a Mexican public hospital". The authors address an important topic and make important contributions to the scientific world.

---

## Round 0.2 · accepted · Accept

Thank you for your revised manuscript. As you can see, our reviewers had no further questions and I am delighted to accept your work.

There are two small points that should be addressed when preparing the final version for publication, neither of which warrants another round of revisions though.

On line 153, when you say “Groups were formed according to the answers to questions 17 and 18.”, I suggest adding a reference to Supplementary file A so that readers will go there to find the questions rather than look at the previous text (where I can’t see any information about these question numbers).

Lines 210–211: There is repetition of text in “but there was no significant there was no evidence of a difference between the groups on all other questions” which must be corrected.

Reviewer 1 ·

Basic reporting

Authors have made improvements.

Experimental design

no comment

Validity of the findings

no comment

Additional comments

no comment